# Immune Durability and Breakthrough Infections 15 Months After SARS-CoV-2 Boosters in People over 65: The IMMERSION Study

**DOI:** 10.3390/vaccines13070738

**Published:** 2025-07-09

**Authors:** Concepció Violán, Bibiana Quirant-Sánchez, Maria Palau-Antoja, Dolors Palacin, Edwards Pradenas, Macedonia Trigueros, Guillem Pera, Gemma Molist, Gema Fernández-Rivas, Marc Boigués, Mar Isnard, Nuria Prat, Meritxell Carmona-Cervelló, Noemi Lamonja-Vicente, Brenda Biaani León-Gómez, Eva María Martínez-Cáceres, Pere Joan Cardona, Julià Blanco, Marta Massanella, Pere Torán-Monserrat

**Affiliations:** 1Unitat de Suport a la Recerca Metropolitana Nord, Institut Universitari d’Investigació en Atenció Primària Jordi Gol (IDIAP Jordi Gol), Mare de Déu de Guadalupe 2, Planta 1ª, Mataró, 08303 Barcelona, Spain; mariapalauantoja@gmail.com (M.P.-A.); mpalacin.centre.ics@gencat.cat (D.P.); gpera@idiapjgol.info (G.P.); gemma.molist@umedicina.cat (G.M.); mcarmonace.mn.ics@gencat.cat (M.C.-C.); nlamonjav.mn.ics@gencat.cat (N.L.-V.); bleongo.mn.ics@gencat.cat (B.B.L.-G.); ptoran.bnm.ics@gencat.cat (P.T.-M.); 2Germans Trias i Pujol Research Institute (IGTP), 08916 Badalona, Spain; bquirant.germanstrias@gencat.cat (B.Q.-S.); mboigues.germanstrias@gencat.cat (M.B.); emmartinez.germanstrias@gencat.cat (E.M.M.-C.); jblanco@irsicaixa.es (J.B.); 3Grup de Recerca en Impacte de les Malalties Cròniques i les seves Trajectòries (2021-SGR-01537), IDIAPJGol, 08007 Barcelona, Spain; 4Red de Investigación en Cronicidad, Atención Primaria y Prevención y Promoción de la Salud (RICCAPPS), Instituto de Salud Carlos III, 08196 Madrid, Spain; 5Immunology Department, Federation of Clinical Immunology Societies Center of Excellence, Universitat Autònoma de Barcelona, 08913 Cerdanyola del Vallès, Spain; 6Laboratori Clinic Metropolitana Nord, Immunology Division, Hospital Universitari Germans Trias i Pujol, 08916 Badalona, Spain; 7IrsiCaixa, Carretera de Canyet, s/n, 08916 Badalona, Spain; epradenas@irsicaixa.es (E.P.); mtrigueros@irsicaixa.es (M.T.);; 8Faculty of Medicine, University of Vic-Central University of Vic (UVic-UCC), Catalonia, 08500 Vic, Spain; 9Clinical Laboratory North Metropolitan Area, Microbiology Department, Germans Trias i Pujol University Hospital, 08916 Badalona, Spain; gfernandezr.germanstrias@gencat.cat (G.F.-R.); pcardonai.germanstrias@gencat.cat (P.J.C.); 10Department of Genetics and Microbiology, Autonomous University of Barcelona, 08916 Badalona, Spain; 11Direcció d’atenció Primària Metropolitana Nord, Institut Català de la Salut, Sabadell, 08204 Barcelona, Spain; marisnard_ext@catsalut.cat (M.I.); nprat@gencat.cat (N.P.); 12Centro de Investigación Biomédica en Red de Enfermedades Infecciosas, Instituto de Salud Carlos III, 28029 Madrid, Spain; 13Multidisciplinary Research Group in Health and Society GREMSAS (2017 SGR 917), 08007 Barcelona, Spain; 14Faculty of Medicine, University of Girona, 17071 Girona, Spain

**Keywords:** SARS-CoV-2, humoral immunity, cohort, ageing, primary health care

## Abstract

**Background**: SARS-CoV-2 booster vaccination remains essential to prevent severe COVID-19, particularly in vulnerable populations such as older adults. This study evaluated the durability and dynamics of immune responses following booster vaccination(s) in >65-year-old individuals and examined their association with protection against new infections. **Methods**: Immune responses were evaluated at 3, 9, and 15 months post-booster, measuring SARS-CoV-2-specific IgG antibodies against spike [IgG(S)] and nucleocapsid [IgG(N)] proteins, neutralizing activity against the Omicron BA.2 variant, and cellular immunity. A subset of participants was tested before booster administration. Regression analyses examined the influence of clinical and immunological factors—including a bivalent fourth dose—on infection risk over time. **Results**: Booster vaccination significantly enhanced IgG(S) and neutralizing capacity, peaking at 3 months. Although a decline was observed by 9 months, responses remained above baseline. Individuals with prior SARS-CoV-2 infection exhibited higher IgG(S) levels and neutralizing titers, and significantly lower reinfection rates (15%), compared to uninfected individuals. A fourth vaccine dose further increased IgG(S) levels. While neutralizing capacity was not consistently enhanced by the fourth dose, recipients experienced a lower rate of new infections. Immune trajectory analyses revealed that breakthrough infections elicited strong humoral responses comparable to those seen in previously infected individuals, highlighting the role of hybrid immunity. **Conclusions**: In older adults, booster vaccination induces durable immune responses, with hybrid immunity offering enhanced protection. A fourth dose boosts antibody levels and reduces infection risk, supporting its use in this high-risk group. Continued monitoring is needed to determine the long-term effectiveness of boosters, particularly against emerging variants.

## 1. Introduction

The COVID-19 pandemic continues to evolve, with the Omicron variant and its sub-lineages now accounting for the majority of global infections [1]. Although Omicron infection is generally associated with lower hospitalization and mortality rates compared to earlier variants, its high transmissibility remains a major concern, particularly for individuals over 65, who face increased risk of severe disease and complications [2]. For this vulnerable group, robust and durable vaccine-induced immunity is essential to ensure sustained protection against emerging variants [3,4].

Long-term vaccine-induced immunity is crucial for defending against rapidly evolving SARS-CoV-2 strains. Studies in at-risk groups, including nursing home residents, have shown a significant increase in plasma neutralizing capacity after the second dose of mRNA COVID-19 vaccines [5]. However, other studies have reported a decline in antibody levels and T-cell responses over time following booster doses, with a 1-log reduction in neutralizing antibody titers within six months [6]. Previous studies revealed that older adults have lower neutralizing antibodies than younger individuals even after two doses of BNT162b2 [7,8]. Notably, despite lower initial antibody responses, older adults can achieve robust immunity with additional booster doses, underscoring the importance of continued booster vaccines in this population [7]. The underlying causes of some individuals being low or non-responders to vaccination remain unclear and cannot be fully explained by age, comorbidities, or baseline immune status.

Booster doses have proven essential for maintaining protective immunity, particularly against rapidly evolving variants, such as Omicron. While these booster doses elicit a significant increase in antibody and T-cell responses, these responses tend to wane over time, especially in older adults. The decline in neutralizing antibody titers within months of vaccination raises important concerns about the durability of protection in the older population [9,10]. However, subsequent booster doses have been shown to restore antibody levels and enhance immune memory, offering protection against infection and severe outcomes despite Omicron’s immune-evasive properties [4].

Among the various factors influencing immune durability, the type of vaccine platform used plays a critical role in shaping both the magnitude and longevity of the immune response. SARS-CoV-2 vaccines have been developed using diverse platforms, and those administered in Spain included mRNA-based (BNT162b2, Pfizer-BioNTech, Mainz, Germany, and mRNA-1273, Moderna, Cambridge, MA, USA) and viral-vector-based (AZD1222, AstraZeneca, Cambridge, United Kingdom and Ad26.COV2.S, Janssen, Leiden, The Netherlands) platforms. Each platform elicits distinct immunological profiles, with mRNA vaccines generally inducing higher neutralizing antibody titers and more robust T-cell responses compared to other platforms [11,12]. However, these responses may wane more rapidly, particularly in older adults. Understanding how these platforms perform in older populations is essential for optimizing booster strategies and tailoring vaccine recommendations to maximize long-term protection.

In response to the increased incidence of breakthrough infections among vaccinated individuals and the observed decline in antibody levels over time, public health agencies including the U.S. Centers for Disease Control and Prevention and the European Medicines Agency recommended a fourth vaccine dose for older adults [13,14,15,16,17]. Studies have shown that a third vaccine dose significantly increases antibody levels, although much of the available data focus on healthier populations [18,19]. A cross-sectional study in nursing home residents demonstrated improved neutralization against the wild-type virus after the third vaccine dose [19].

While previous research has examined immune responses following three vaccine doses in older adults, including both previously infected and uninfected adults [20], data on immune responses in those who subsequently experience breakthrough infections remain limited. Emerging evidence highlights the significant benefits of hybrid immunity—a combination of infection and vaccination—which offers superior and longer-lasting protection than vaccination alone. This is particularly relevant for high-risk groups, such as older individuals in communal or nursing home settings [21].

A deeper understanding of the full immune response, including both humoral and cellular immunity, in older adults living in the community, is key to shaping future public health strategies. It is also critical to determine whether current vaccines, based on the ancestral variant, remain effective against newer immune-evasive variants. Moreover, evaluating the impact of comorbidities and polypharmacy on immunity will help to refine health policy recommendations for this population [17,22,23].

This study aims to investigate the association between SARS-CoV-2-specific antibodies—IgG(S) and neutralizing antibodies—and protection against new infections (including reinfections and breakthrough cases) 15 months after booster vaccination in older adults, adjusting for age, sex, prior infection, vaccine type, and other relevant clinical variables.

## 2. Materials and Methods

### 2.1. Study Design

The IMMERSION Study is a prospective study conducted in the Northern Metropolitan Area of Barcelona (Spain). The study involved two groups of older adults attending primary care: those with and without prior SARS-CoV-2 infection (infected and uninfected). The ethics committee of the Foundation University Institute for Primary Health Care Research Jordi Gol i Gurina approved the study protocol (ref. 21/254-PCV).

### 2.2. Participant Recruitment, Follow-Up

A total of 287 individuals aged 63 years and older were enrolled through primary health centers between 13 December 2021 and 7 March 2023. Inclusion criteria required informed consent and availability for follow-up visits at 3, 9, 15 months. Among the participants, 86 were recruited before the third dose (13 December 2021–24 January 2022), while 201 participants were recruited three months post-third vaccine dose (31 January–10 March 2022). At the 15-month visit, participants were further stratified into two groups: those who had received a fourth vaccine dose (N = 160, 3 months post-fourth booster dose) and those who had not (N = 73). Participants were categorized as infected or uninfected based on RT-PCR and IgG(N) antibody tests at the first visit; a positive result in either test indicated prior infection. Due to the retrospective nature of prior infection reporting, precise dates of primary infections were not available for most participants, either because infections were asymptomatic or not clinically recorded. As such, time intervals between infection and vaccination or between reinfections could not be reliably calculated. Five participants who did not receive all three scheduled vaccine doses were excluded (Figure 1).

### 2.3. Study Outcomes

The primary outcomes of the study were breakthrough infection, defined as SARS-CoV-2 infection ≥15 days after the third dose in individuals without prior infection, and reinfection, defined as a new infection ≥90 days after a prior infection confirmed by RT-PCR or IgG(N) serology positivity (>11.5 BAU/mL). This distinction was made to reflect distinct immunological contexts: breakthrough infections occur in infection-naïve vaccinated individuals, while reinfections represent a secondary exposure in individuals with hybrid immunity [24,25,26,27].

### 2.4. Covariables

At enrolment, participants completed clinical questionnaires covering COVID-19-specific symptoms, comorbidities, current medications, nutrition, and vaccination history. For comparison purposes, participants were grouped by age (63–74, 75–84, and ≥85 years), comorbidity burden (0, 1–2, 3–5, 6–9, and >9 active chronic conditions as of 1 January 2021), and vaccination strategy (homologous: three doses of mRNA vaccines: BNT162b2 or mRNA-1273; or heterologous: vector-based vaccines such as AZD1222 or Ad26.COV2.S followed by an mRNA vaccine). The third booster mRNA vaccines targeted the original WH-1 spike protein, while the fourth boosters used bivalent mRNA vaccines covering the WH1 strain and Omicron BA.1 sub-lineage.

### 2.5. Determination of Total Anti-SARS-CoV-2 Antibodies, Neutralization Titers, and T-Cell Responses

Serum levels of anti-spike [IgG(S)] and anti-nucleocapsid IgG [IgG(N)] antibodies were quantified using standardized immunoassays, and serum neutralizing capacity was assessed via a pseudovirus-based assay against the Omicron BA.2 variant [28,29,30]. Cellular immunity was evaluated in a subset of participants (N = 31) using a whole-blood stimulation assay measuring IFN-γ production in response to spike protein peptides (see Section A.1 for more details).

### 2.6. Determination of Different Trajectories According to Infected or Uninfected Status at Baseline

Participants were classified into eight trajectory groups based on their SARS-CoV-2 infection status over time (see Section A.2). Briefly, Trajectory 1 (T1) included individuals who remained uninfected throughout the entire study period. Trajectory 2 (T2) comprised uninfected participants who experienced a breakthrough infection between 9 and 15 months post-booster, while Trajectory 3 (T3) included those who became infected between 3 and 9 months after the third dose. Trajectory 4 (T4), defined as uninfected individuals who developed a breakthrough infection between 3 and 9 months post-booster, was excluded from the main analysis due to limited sample size (n < 10). Trajectory 5 (T5) consisted of participants who were already infected at baseline and remained positive at all subsequent visits, with no evidence of reinfection. Additional trajectories—Trajectory 4 (uninfected at baseline with breakthrough infection between 3 and 9 months) and Trajectories 6 to 8 (various reinfection patterns)—were excluded from the main analysis due to limited sample size (n < 10), which restricted statistical power. Participants were also stratified according to vaccination regimen: those who received only three doses, and those who received a fourth vaccine dose at 9 months. For the purposes of trajectory analysis at early time points (baseline, 3 months, and 9 months), individuals from both the three- and four-dose groups were pooled, as none had received the fourth dose by that point. Merging these groups increased statistical power and allowed for more robust comparison across immune trajectories.

### 2.7. Sample Size and Statistical Power

To detect a mean difference of 0.35 points between two groups (previously infected versus previously uninfected individuals), with a standard deviation of 1 point, expecting twice as many non-infected participants as infected patients, a sample of 291 subjects is required, with a power of 80% and a significance of 5% with a bilateral test.

### 2.8. Statistical Analysis

Descriptive analysis used frequencies and percentages for categorical variables and mean (SD) or median [IQR] for quantitative variables. Group comparisons were assessed using the Mann–Whitney U test for unpaired data and the Wilcoxon signed-rank test for paired longitudinal comparisons. Logistic regression models were used to assess the association between the outcome variables (reinfection and breakthrough) and independent variables, including vaccination type (homologous, heterologous), prior infection status, number of vaccinations, chronic disease count, age, and sex. For regression models, antibody levels were log transformed to correct for skewness. Univariate analyses estimated unadjusted associations, with variables showing statistical significance (*p* < 0.05) included in the multivariate model.

The multivariate analysis was adjusted for potential confounders, with age and sex included in all models regardless of univariate significance. Adjusted odds ratios (ORs) with 95% confidence intervals (CIs) were reported.

All tests were two-sided, with statistical significance set at *p* < 0.05. Analyses were performed using R version 4.3.3.

## 3. Results

### 3.1. Participant Characteristics

This study included 287 participants (Figure 1), 53% of whom were female. Participants’ characteristics at each visit are shown in Table 1. No major sex-based differences were observed in antibody levels or infection rates across visits, supporting the comparability of groups; therefore, results are presented together for men and women in the subsequent analyses.

The median age at enrolment was 69 years in the baseline sub-cohort, which was younger than the follow-up cohort (median 76 years, Appendix A). Among all participants, 86 were recruited before their third SARS-CoV-2 vaccine dose, while 233 completed all follow-ups—160 [68.7%] received four vaccine doses and 73 [31.3%] received only three doses at month 15 (Appendix A).

IgG(S) and IgG(N) antibody levels and neutralization (NAb) titers were similar between sexes across all time points. All individuals over 75 received homologous vaccination, while 39.2% of those aged 63–74 received heterologous vaccination (Appendix A). At recruitment, 69.3% of participants were uninfected, but by the end of the study, only 22.7% remained uninfected. Most infections were identified through serological testing (Appendix A).

### 3.2. Impact of Prior Infection on Immunity After the Third Vaccine Dose

To assess the impact of prior infection, we examined infection rates following the third dose. From baseline to visit 3, 12.5% of uninfected participants developed breakthrough infection, versus 4.5% of those previously infected (*p* = 0.44). This trend became more pronounced in the longer-term follow-up: between visit 3 and visit 9, 44.9% of previously uninfected individuals were infected, compared to only 6.6% of those with prior infection (*p* = 0.0004). These findings support analyzing breakthrough and reinfection cases separately, as they reflect distinct susceptibility profiles. Although the number of reinfection cases was limited, they provide valuable insight into the durability of hybrid immunity in older adults. We also evaluated immune responses in a combined group (reinfections and breakthrough infections [R + BT]) to explore the cumulative effect of repeated exposures.

IgG(N) levels—used to confirm prior SARS-CoV-2 infection—remained undetectable in uninfected individuals throughout the study, consistently supporting their seronegative status (Figure 2A).

Participants with a history of infection exhibited significantly higher IgG(N) levels at all visits (*p* < 0.0001), and reinfected and breakthrough cases showed similar levels (Figure 2A). At 15 months, stratification by vaccine doses revealed no significant differences in IgG(N) levels between participants who received three or four doses within each infection status (Figure 3A).

Uninfected individuals showed lower IgG(S) levels than those with prior infection throughout the study (Figure 2B). At 9 months, participants with either breakthrough or prior infections exhibited significantly higher IgG(S) titers (3.85 and 3.56 log_10_ BAU/mL) compared to uninfected individuals (3.09 log_10_ BAU/mL, *p* < 0.0001, Figure 2B). At 15 months post-third dose, IgG(S) titers remained higher in previously infected, reinfected, and breakthrough groups compared to uninfected individuals, with reinfected participants showing the highest levels (3.89 log_10_ BAU/mL). When considering both three- and four-dose recipients at month 15, IgG(S) titers were generally higher in the four-dose group across all infection categories (*p* < 0.01 in all cases)—except for reinfected individuals (Figure 3B).

Regarding neutralization capacity, previously infected participants had significantly higher serum neutralization levels than uninfected ones across all time points (*p* < 0.0001 in all cases), except at 15 months (three-dose recipients only, Figure 2C). At 15 months, only the breakthrough group showed significantly higher NAb titers when comparing those who received three vs. four doses (Figure 3C, *p* < 0.001).

To further explore the potential clinical relevance of a fourth vaccine dose, we evaluated the occurrence of new infections between the 9- and 15-month visits. New SARS-CoV-2 infections occurred in 19.7% (14/71) of individuals who had received three vaccine doses, compared to 8.9% (14/158) of those who received four doses. This difference was statistically significant (odds ratio [OR] = 0.40, 95% confidence interval [CI]: 0.18–0.88; *p* = 0.028), indicating that a fourth dose was associated with a significantly lower likelihood of subsequent infection. Due to the small number of reinfection cases during this interval, a specific analysis of reinfections could not be performed.

### 3.3. Vaccine Dose and Immune Response Dynamics (Trajectories)

Both IgG(S) and neutralizing antibody responses were analyzed longitudinally, stratified by immune trajectory and number of vaccine doses received (Table 2 and Figure 4). To enhance statistical power, data from the three- and four-dose groups were merged for months 0, 3, and 9, as the fourth dose was administered only after the 9-month visit (Table 2). To better visualize the effect of vaccine doses on antibody trajectories, responses were also stratified by number of doses (3 doses vs. 4 doses) and plotted separately for each immune trajectory group (Figure 4A,B).

Across all trajectories, the third vaccine dose elicited a significant increase in both IgG(S) and neutralizing antibodies by 3 months post-booster (*p* < 0.02 in all groups). For example, in uninfected individuals (T1), IgG(S) levels rose from 2.26 [1.42–2.69] to 3.31 [3.08–3.47] log_10_ BAU/mL (*p* < 0.001), and neutralizing titers increased from 1.78 to 2.60 log_10_ (*p* < 0.001). However, a marked decline was observed by month 9 in this group, with IgG(S) levels dropping to 3.15 log10 BAU/mL and neutralizing titers decreasing to 2.09 (*p* = 0.02 for both), indicating poor persistence in infection-naïve individuals.

Uninfected participants with a breakthrough infection between months 3 and 9 (T3) demonstrated a robust post-infection increase in both IgG(S) levels and neutralizing titers during this period (*p* < 0.001 in both cases). Their IgG(S) levels rose from 3.29 to 3.86 log_10_ BAU/mL (*p* < 0.001), and neutralizing titers from 2.47 to 3.35 log_10_ (*p* < 0.001), surpassing levels seen in chronically infected individuals (T5) by month 9 (*p* < 0.001 for both measures), illustrating the immunological benefit of early hybrid immunity.

After month 9, immune trajectories began to diverge based on whether participants had received the fourth dose. Among those who received the fourth dose, IgG(S) levels remained high across all trajectories at month 15. For instance, in T1, IgG(S) levels rose to 3.76 [3.39–4.06] log_10_ BAU/mL (*p* < 0.001), whereas in T2, T3, and T5, they remained above 3.9 log_10_ BAU/mL. This suggests a broad boosting effect of the fourth dose on total anti-spike antibodies, independent of infection status.

In contrast, neutralizing antibody responses revealed greater variability. In uninfected individuals (T1), neutralizing titers rose modestly with the fourth dose (from 2.09 to 2.22 log_10_ titer; *p* = 0.008), but in the three-dose group, they continued to decline (*p* = 0.05). In participants where infection had already occurred before the fourth dose (T3), titers at month 15 remained high regardless of a fourth dose (3.33–3.35 log_10_ titer), underscoring the dominant effect of infection in shaping neutralizing immunity. Participants who experienced a late breakthrough infection (T2) between months 9 and 15 also showed marked increases in IgG(S) and neutralizing titers at month 15 (*p* = 0.008 in both), reaching levels comparable to those of T5, thus reinforcing the added value of hybrid immunity, even at later stages.

### 3.4. Risk of Reinfection and Breakthrough Infections

Regression models showed a cumulative infection incidence of 48.5% between 3 and 9 months and 68.5% at 3–15 months post-third dose. Among previously uninfected individuals, the incidence at 15 months was 58.9%. In contrast, participants with prior infection had lower reinfection risks: 6.0% at 3–9 months, 5.1% at 3–15 months, and 15.0% overall up to 15 months post-vaccination. All regression models were adjusted for age, sex, IgG(S), NAb titers, and prior infection status. Three models were constructed: two from 3–9 and 3–15 months for the global cohort, and one from baseline to 15 months for the baseline sub-cohort. Prior infection was consistently associated with a reduced risk of reinfection: OR = 0.08 95%CI [0.02; 0.21], OR = 0.04 95%CI [0.01; 0.12], and OR = 0.05 95%CI [0.00; 0.28], respectively (Table 3).

### 3.5. Cellular Immune Response Differences by Sex

Both sexes exhibited positive cellular immune responses following stimulation; however, females exhibited consistently higher IFN-γ levels (Table 4). At three months post-third dose, median IFN-γ responses were 90.0 pg/mL in females vs. 42.8 pg/mL in males, a trend sustained across all follow-ups.

## 4. Discussion

The emergence of SARS-CoV-2 variants, particularly Omicron, has highlighted the need for a comprehensive understanding of immune responses following infection and vaccination, especially in older populations. In this prospective study, we assessed IgG(N), IgG(S), and NAb titers up to 15 months after the third vaccine dose, across different infection statuses, immune response dynamics (trajectories), and vaccination statuses. Our results show that although antibody levels remained above assay-defined positivity thresholds, they were not sufficient to prevent infection in uninfected individuals.

A substantial proportion of these participants developed breakthrough infections during follow-up, particularly between visit 3 and visit 9, when 44.9% of previously uninfected individuals became infected despite prior vaccination. In contrast, individuals who had already been infected before the third dose showed markedly lower reinfection rates—only 6.6% during the same period—suggesting stronger and more durable immune protection in this group. Although the number of reinfections was limited, these findings support the notion that infection-primed immunity offers greater resistance to subsequent exposure and justify analyzing breakthrough and reinfection cases as distinct trajectories.

Participants with prior SARS-CoV-2 infection consistently exhibited significantly higher IgG(N), IgG(S), and NAb titers than uninfected individuals, supporting the concept of hybrid immunity [31]. These findings align with previous reports showing that natural infection or reinfection, when combined with vaccination, enhances immune responses and durability in older adults [32]. In our study, both reinfected and breakthrough groups showed sustained IgG(S) and NAb levels, with reinfected individuals displaying the highest IgG(S) titers at 15 months.

Reinfected individuals exhibited higher IgG(N), IgG(S), and NAb titers than those with a single prior infection, but the differences were not statistically significant, likely due to the limited sample size. In contrast, uninfected individuals consistently exhibited lower antibody levels throughout the follow-up. This observation aligns with previous studies showing that lower and more rapidly waning post-vaccination titers in uninfected individuals correlate with a higher risk of reinfection, particularly with Omicron waves [33]. Furthermore, the severity of the initial infection has been linked to antibody durability, with more severe cases maintaining higher NAb levels over time [34,35]. In our cohort, most infections were mild, which may partially explain the moderate and variable antibody responses observed. Intrinsic immune variability may also contribute to limited humoral memory in some individuals, even after repeated exposures [35].

Our neutralization assays were performed using a BA.2-based pseudovirus, while the fourth vaccine dose administered during the study was a bivalent mRNA vaccine targeting the ancestral WH-1 strain and Omicron BA.1 sub-lineage. This mismatch between the vaccine composition and the circulating BA.2 variant may explain why we did not observe significant differences in NAb titers between uninfected and infected participants who received three or four doses. In contrast, among individuals with breakthrough infections—likely caused by the BA.2 variant—those who received a fourth dose exhibited significantly higher NAb titers compared to those with three doses, suggesting that the combination of infection and vaccination (i.e., hybrid immunity) drove the enhanced neutralizing response, rather than the additional booster alone. This also helps explain why IgG(S) and IgG(N) levels—measured using commercial assays based on the ancestral strain—showed clearer differences between groups: these assays capture the cumulative exposure to SARS-CoV-2 antigens (through infection or vaccination), while neutralization assays reflect functional immunity against specific variants like BA.2, which may not be effectively targeted by ancestral strain-based vaccines alone.

Our trajectory-based analysis further illustrates how the timing of infection and additional vaccination events influence the magnitude and durability of antibody responses. Following the third vaccine dose, all participants experienced a peak in IgG(S) and neutralizing titers by 3 months, followed by a marked decline at 9 months, particularly among uninfected individuals. This shared early trajectory highlights the consistent but transient impact of the third dose. After month 9, immune trajectories began to diverge. Individuals who received a fourth dose maintained high IgG(S) levels across all infection statuses, suggesting a strong boosting effect on total spike-specific antibodies. However, neutralizing responses remained more dependent on infection timing. Notably, individuals with a breakthrough infection between months 3–9 (T3) showed robust neutralizing titers at month 15, similar to those chronically infected (T5), while the fourth dose appeared to have a limited additional effect on neutralization in this group. In contrast, participants who experienced late breakthrough infections (T2) exhibited a sharp increase in both IgG(S) and NAb titers after month 9, underscoring the strength of hybrid immunity even when infection occurred after the fourth dose. These findings reinforce the importance of both infection history and vaccine timing in shaping durable immune protection in older adults.

While prior infection clearly reduced the risk of reinfection, age, sex, IgG(S), and NAb levels alone did not reliably predict protection. Some individuals may generate lower antibody levels despite multiple exposures, and mild infections may not induce long-lasting humoral immunity. These observations highlight the role of host-specific factors, particularly immunosenescence, which affects all arms of the immune system, weakening both immune responses to infection and the development of long-term immune memory post-vaccination [36]. Notably, in our study, heterologous versus homologous vaccination types did not result in significant differences in IgG(S) levels or neutralization titers in the univariate model. However, despite age-related immune decline, vaccination (alone or combined with prior infection) proved effective in preventing severe COVID-19 outcomes in this older population.

Notably, despite the high prevalence of multimorbidity in our cohort (88.3%), prior infection remained protective, while antibody levels did not correlate with reinfection risk [36,37,38]. Although we assessed comorbidity burden numerically, we recognize that not all conditions have the same impact on immune function. Detailed data on disease types and medication use were collected, but in-depth analysis was beyond the scope of this study. Future work should explore the role of specific comorbidities and treatments in shaping immune responses in older adults.

We also observed stronger cellular immune responses in females, consistent with previously reported sex-based differences linked to genetic and hormonal factors [39]. Importantly, no participants required COVID-19-related hospitalization, suggesting that vaccination, with or without prior infection, provided effective protection against severe disease during the Omicron wave. While our study did not include younger individuals, previous research has shown that aging is associated with a decline in IFN-γ production, particularly affecting T-helper cell function and vaccine responsiveness [40,41]. Our findings suggest that sex may further modulate this age-related immune decline.

In addition to immunological outcomes, we evaluated the clinical impact of the fourth vaccine dose by assessing the incidence of new SARS-CoV-2 infections between 9 and 15 months post-vaccination. Individuals who received four doses had a notably lower rate of new infections compared to those with only three doses. This finding highlights the added value of continued booster vaccination in preventing infections in older adults, a population that remains at increased risk for severe COVID-19, hospitalization, and mortality. Reducing infection rates in this vulnerable group is essential for minimizing healthcare burden and promoting healthy aging.

A major strength of this study lies in its prospective design and extended follow-up, which allowed for the longitudinal assessment of antibody responses in a real-world aging population. The repeated measurement of IgG(S) and neutralizing antibodies using sensitive assays, combined with high participant retention and careful adjustment for confounding variables, adds robustness to our findings. Nonetheless, some limitations should be acknowledged. The lack of universally accepted protective antibody thresholds complicates the interpretation of immune correlates of protection. Moreover, neutralization assays focused solely on the Omicron BA.2 variant, limiting broader generalizability to other strains. Also, cellular immunity analyses were conducted in a relatively small subgroup, which may not fully capture the diversity of T-cell responses in this population. A limitation of our study is the lack of accurate data on the timing of primary infections. Many infections occurred asymptomatically or were unreported, and dates were not consistently documented in clinical records. This limited our ability to assess associations between time since infection and immune response. Finally, the low number of individuals in some comparison groups (i.e., due to the timing and occurrence of reinfections or breakthrough cases) may limit statistical power and complicate interpretation of between-group differences.

## 5. Conclusions

In summary, the study highlights the complexity of the immune response to SARS-CoV-2 in older adults. Prior infection significantly boosts both IgG(S) levels and NAb titers against Omicron following booster vaccination, supporting the role of hybrid immunity in achieving stronger and more durable protection. The third booster dose elicited peak NAb titers at 3 months, particularly among uninfected individuals. However, NAb levels declined substantially over time, and remained low at 15 months, even after a fourth dose, which did not significantly enhance neutralization against Omicron, likely due to the mismatch between vaccine formulation and circulating variants. In contrast, breakthrough infections likely involving BA.2 led to stronger neutralizing responses. Despite antibody waning, prior infection conferred significant protection against reinfection, highlighting the importance of immune memory beyond humoral markers.

These findings have important implications for vaccination strategies in older adults. They support continued immunization for those over 65 and underline the added benefit of hybrid immunity. In addition, our analysis shows that a fourth vaccine dose was associated with a significantly reduced risk of new infections during the follow-up period, reinforcing its role in infection prevention even amid variant mismatch and waning antibody levels. The results also emphasize the need for variant-adapted vaccines and ongoing immune monitoring, particularly considering immunosenescence and evolving variants.

## Figures and Tables

**Figure 1 vaccines-13-00738-f001:**
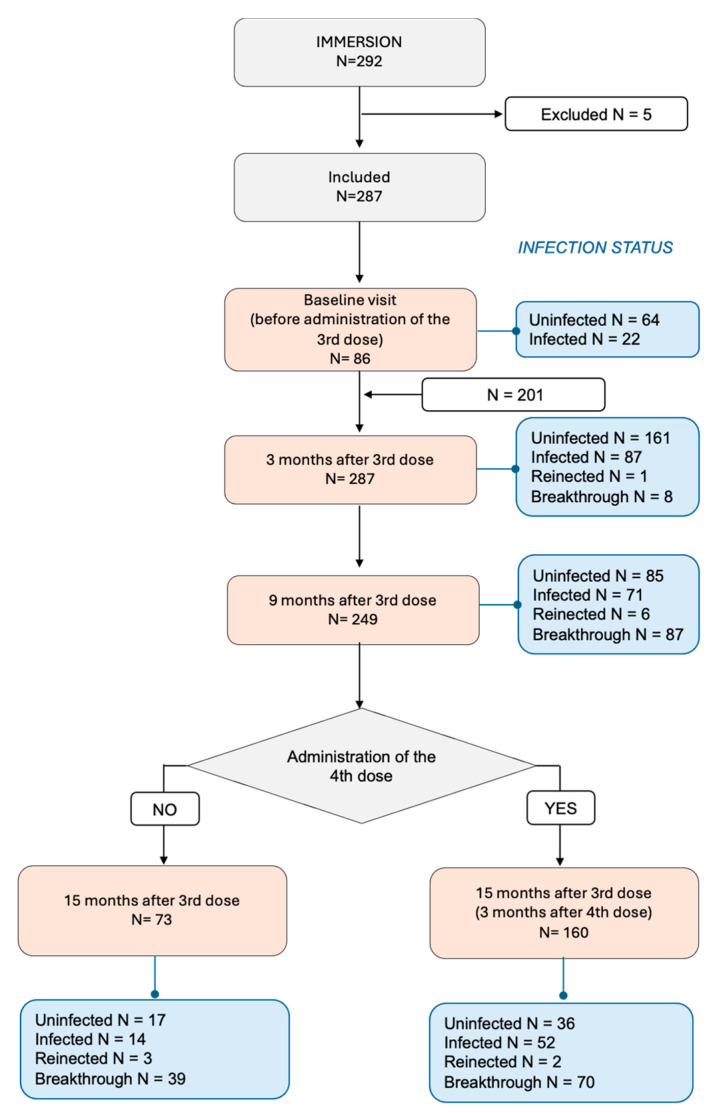
Study population flowchart. The number of subjects included in each visit and their infection status are noted.

**Figure 2 vaccines-13-00738-f002:**
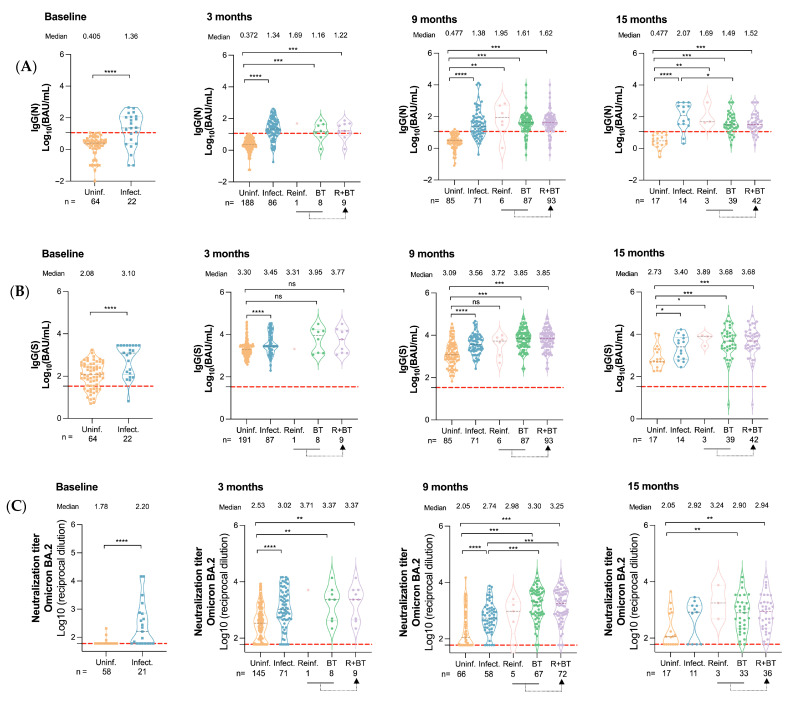
Comparison of IgG(N), IgG(S) levels, and neutralizing activity between uninfected, infected, breakthrough, and reinfected individuals before and 3, 9, and 15 months after the administration of the third dose of the SARS-CoV-2 vaccine. Antibody levels are represented with a violin plot, with individual data points overlaid. The number inside the square indicates the median value for each group. The dashed line represents the positivity threshold. (**A**) Levels of IgG(N) by infection status across all time points. (**B**) Levels of IgG(S) by infection status across all time points. (**C**) Levels of neutralizing antibodies by infection status across all time points. At 15 months, only the individuals who did not receive the fourth dose are included. *p*-values were obtained using the Wilcoxon test, taking the uninfected group as the reference group. Significance levels were reported as follows: ns for *p*-value > 0.05; * for *p*-value ≤ 0.05; ** for *p*-value ≤ 0.01; *** for *p*-value ≤ 0.001; and **** for *p*-value ≤ 0.0001. Group abbreviations: Uninf. refers to uninfected individuals; Infect. to those previously infected; Reinf. to reinfected individuals; BT indicates breakthrough infections; R + BT denotes individuals who experienced either reinfection or breakthrough infection, representing cumulative hybrid exposure.

**Figure 3 vaccines-13-00738-f003:**
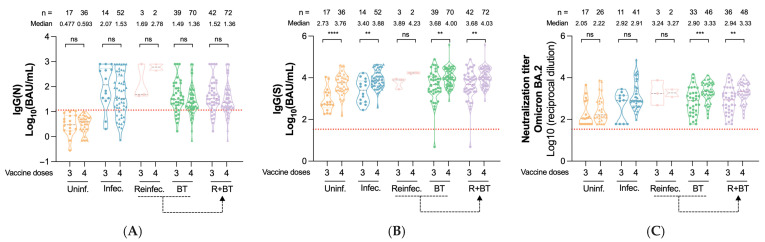
Comparison of IgG(N) levels, IgG(S) levels, and neutralizing activity between individuals with three and four doses of the SARS-CoV-2 vaccine at 15 months regarding their infection status. Antibody levels are represented with a violin plot, with individual data points overlaid. The number inside the square indicates the median value for each group. The dashed line represents the positivity threshold. (**A**) Levels of IgG(N) 15 months after the administration of the third dose of the SARS-CoV-2 vaccine. (**B**) Levels of IgG(S) 15 months after the administration of the third dose of the SARS-CoV-2 vaccine. (**C**) Levels of neutralizing antibodies 15 months after the administration of the third dose of the SARS-CoV-2 vaccine. *p*-values were obtained using the Wilcoxon test. Significance levels were reported as follows: ns for *p*-value > 0.05; ** for *p*-value  ≤  0.01; *** for *p*-value  ≤  0.001; and **** for *p*-value  ≤  0.0001. Group abbreviations: Uninf. refers to uninfected individuals; Infect. to those previously infected; Reinf. to reinfected individuals; BT indicates breakthrough infections; R + BT denotes individuals who experienced either reinfection or breakthrough infection, representing cumulative hybrid exposure.

**Figure 4 vaccines-13-00738-f004:**
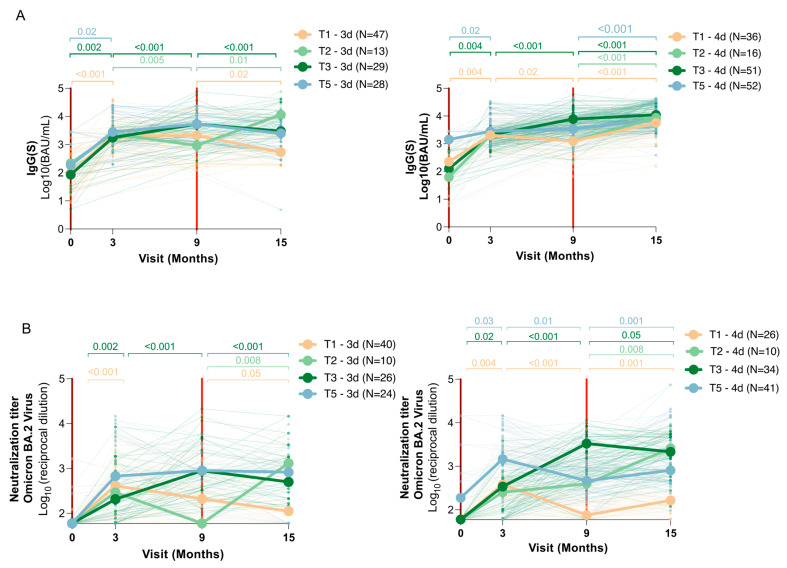
Longitudinal evolution of antibody responses in older adults following SARS-CoV-2 vaccination, stratified by infection trajectory and vaccine dosing. Levels of IgG(S) (**A**) and neutralizing antibodies against Omicron BA.2 (**B**) are shown at baseline (V0) and at 3, 9, and 15 months after administration of the third vaccine dose. Participants were categorized into four immune trajectories based on infection timing: T1 (uninfected throughout), T2 (breakthrough infection between months 9 and 15), T3 (breakthrough infection between months 3 and 9), and T5 (infected at baseline with no reinfection). Median values are plotted. The red line indicates when the third (baseline) and the fourth (month 9) doses were administered. From baseline to month 9, trajectories reflect the effect of the third dose alone; divergence between groups at month 15 reflects the additional effects of infection or receipt of a fourth dose. *p*-values were obtained using the Wilcoxon test.

**Table 1 vaccines-13-00738-t001:** Descriptive study population with the serological response, infection and vaccination status, and sociodemographic and clinical characteristics regarding the participants’ sex.

Sex	Baseline (N = 86)		3-Month Visit (N = 287)		9-Month Visit (N = 249)		15-Month Visit (N = 233)	
	Female (n = 52)	Male (n = 34)	Female (n = 152)	Male (n = 135)	Female (n = 134)	Male (n = 115)	Female (n = 123)	Male (n = 110)
**Sociodemographic characteristics**								
Age, years, *median* [Q1; Q3]	69.0 [67.0; 76.0]	68.0 [66.0; 73.8]	75.0 [70.0; 81.2]	77.0 [72.5; 84.0]	75.0 [70.0; 81.0]	77.0 [72.0; 83.0]	74.0 [70.0; 80.0]	76.0 [72.0; 82.0]
Age, years (categorical), *n* (%)								
63–74	37 (71.2)	27 (79.4)	71 (46.7)	49 (36.3)	65 (48.5)	45 (39.1)	64 (52.0)	45 (40.9)
75–84	13 (15.0)	5 (14.7)	60 (39.5)	61 (45.2)	51 (38.1)	50 (43.5)	49 (39.8)	48 (43.6)
>84	2 (3.8)	2 (5.9)	21 (13.8)	25 (18.5)	18 (13.4)	20 (17.4)	10 (8.1)	17 (15.5)
**Health problems ^1^**								
Number of diseases, *n* (%)								
0	1 (1.9)	1 (3.0)	1 (0.7)	2 (1.5)	1 (0.8)	2 (1.8)	1 (0.8)	2 (1.9)
1–2	5 (9.8)	7 (21.2)	13 (8.7)	17 (12.9)	11 (8.4)	16 (14.3)	10 (8.3)	16 (14.9)
3–5	13 (25.5)	4 (12.1)	31 (20.8)	22 (16.7)	26 (19.8)	17 (15.2)	24 (20.0)	18 (16.8)
6–9	11 (21.6)	11 (33.3)	37 (24.8)	45 (34.1)	34 (26.0)	39 (34.8)	33 (27.5)	34 (31.8)
>9	21 (41.2)	10 (30.3)	67 (45.0)	46 (34.8)	59 (45.0)	38 (33.9)	52 (43.3)	37 (34.6)
**Vaccination status**								
Number of vaccine doses, *n* (%)								
2	52 (100)	34 (100)	.	.	.	.	.	.
3	.	.	152 (100)	135 (100)	134 (100)	115 (100)	42 (34.1)	31 (28.1)
4	.	.	.	.	.	.	81 (65.9)	79 (71.8)
Vaccination strategy ^2^, *n* (%)								
Heterologous	27 (51.9)	19 (55.9)	27 (17.8)	20 (14.8)	25 (18.7)	18 (15.7)	25 (20.3)	18 (16.4)
Homologous	25 (48.1)	15 (44.1)	125 (82.2)	115 (85.2)	109 (81.3)	97 (84.3)	98 (79.7)	92 (83.6)
**SARS-CoV-2 infection**								
Infection status, *n* (%)								
Uninfected	43 (82.7)	21 (61.8)	108 (71.1)	83 (61.5)	49 (36.6)	36 (31.3)	30 (24.4)	23 (20.9)
Infected	9 (17.3)	13 (38.2)	37 (24.3)	50 (37.0)	29 (21.6)	42 (36.5)	27 (22.0)	39 (35.5)
Reinfected	0 (0)	0 (0)	1 (0.7)	0 (0)	4 (3.0)	2 (1.7)	3 (2.4)	2 (1.8)
Breakthrough	0 (0)	0 (0)	6 (3.9)	2 (1.5)	52 (38.8)	35 (30.4)	63 (51.2)	46 (41.8)
**Humoral immune response**								
IgG(N), Log_10_ (BAU/mL), *mean* (SD)	0.33 (0.85)	0.66 (0.97)	0.61 (0.65)	0.77 (0.73)	1.14 (0.91)	1.21 (0.85)	1.27 (0.78)	1.43 (0.82)
IgG(S), Log_10_ (BAU/mL), *mean* (SD)	2.16 (0.72)	2.30 (0.75)	3.39 (0.40)	3.42 (0.47)	3.48 (0.61)	3.54 (0.58)	3.71 (0.67)	3.82 (0.52)
Neutralization capacity ^3^, Log_10_ (reciprocal dilution), *mean* (SD)	1.90 (0.43) *(n = 47)*	2.05 (0.56) *(n = 32)*	2.75 (0.62) *(n = 116)*	2.70 (0.72) *(n = 109)*	2.73 (0.68) *(n = 101)*	2.81 (0.72) *(n = 95)*	2.88 (0.66) *(n = 90)*	2.97 (0.72) *(n = 89)*

^1^ Health problems were not available for six individuals (two from the baseline subgroup). ^2^ The vaccination of subjects who received three doses of an mRNA vaccine (BNT162b2 or mRNA-1273) is counted as a homologous vaccination, while the vaccination of subjects who received at least one dose of a viral vector vaccine (AZD1222 or Ad26.COV2.S) is counted as a heterologous vaccination. ^3^ Neutralizing activity was tested in a total of 225 individuals (79 from the baseline subgroup).

**Table 2 vaccines-13-00738-t002:** Longitudinal IgG(S) and neutralizing antibody titers by immune trajectory group. Trajectory groups were defined by infection status over time. Data from participants who received three and four vaccine doses were merged for visits V0, V3, and V9, as the fourth dose was administered only after month 9.

Trajectory 1 Uninfected in All Visits	IgG(S), Log10 (BAU/mL), Median [IQR]		Neutralization Titers, Log_10_ (Reciprocal Dilution), Median [IQR]	
T1 month 0, all participants	2.260 [1.418, 2.688]	n = 31	1.778 [1.778, 1.778]	n = 26
T1 month 3 all participants	3.307 [3.083, 3.474]	n = 83	2.602 [2.052, 3.081]	n = 66
T1 month 9, all participants	3.146 [2.795, 3.547]	n = 59	2.094 [1.778, 2.866]	n = 48
T1 month 15, 3 doses	2.731 [2.559, 3.293]	n = 17	2.048 [1.778, 2.891]	n = 17
T1 month 15, 4 doses	3.756 [3.393, 4.060]	n = 36	2.215 [2.033, 2.865]	n = 26
*p-values, Wilcoxon*				
T1, Month 0 vs. month 3	*<0.001*	⇧	*<0.001*	⇧
T1, Month 3 vs. month 9	*0.02*	⇩	*0.02*	⇩
T1, Month 9 vs. month 15, 3 doses	*0.02*	⇩	*0.05*	⇩
T1, Month 9 vs. month 15, 4 doses	*<0.001*	⇧	*0.008*	⇧
**Trajectory 2** Breakthrough infection between month 9 and 15	**IgG(S)**, Log10 (BAU/mL), median [IQR]		**Neutralization Titers,** Log_10_ (reciprocal dilution), median [IQR]	
T2 month 0, all participants	1.815 [1.258, 2.444]	n = 7	1.778 [1.778, 1.778]	n = 7
T2 month 3 all participants	3.299 [3.109, 3.451]	n = 29	2.436 [1.975, 2.843]	n = 20
T2 month 9, all participants	2.971 [2.757, 3.459]	n = 27	1.811 [1.778, 2.979]	n = 19
T2 month 15, 3 doses	4.061 [3.681, 4.478]	n = 13	3.117 [2.618, 3.539]	n = 10
T2 month 15, 4 doses	3.938 [3.629, 4.524]	n = 16	3.414 [3.090, 3.880]	n = 9
*p-values, Wilcoxon*				
T2, Month 0 vs. month 3	*0.02*	⇧	*0.02*	⇧
T2, Month 3 vs. month 9	*0.002*	⇩	*0.13*	
T2, Month 9 vs. month 15, 3 doses	*0.01*	⇧	*0.008*	⇧
T2, Month 9 vs. month 15, 4 doses	*<0.001*	⇧	*0.008*	⇧
**Trajectory 3** Breakthrough infection between month 3 and 9	**IgG(S)**, Log10 (BAU/mL), median [IQR]		**Neutralization Titers,** Log_10_ (reciprocal dilution), median [IQR]	
T3 month 0, all participants	1.945 [1.540, 2.134]	n = 19	1.778 [1.778, 1.778]	n = 19
T3 month 3 all participants	3.293 [3.110, 3.447]	n = 80	2.473 [1.981, 2.787]	n = 60
T3 month 9, all participants	3.860 [3.484, 4.119]	n = 80	3.346 [2.869, 3.688]	n = 60
T3 month 15, 3 doses	3.479 [3.307, 3.924]	n = 22	2.700 [2.333, 3.239]	n = 19
T3 month 15, 4 doses	4.045 [3.742, 4.391]	n = 51	3.333 [3.013, 3.673]	n = 34
*p-values, Wilcoxon*				
T3, Month 0 vs. month 3	*<0.001*	⇧	*<0.001*	⇧
T3, Month 3 vs. month 9	*<0.001*	⇧	*<0.001*	⇧
T3, Month 9 vs. month 15, 3 doses	*0.007*	⇩	*<0.001*	⇩
T3, Month 9 vs. month 15, 4 doses	*<0.001*	⇧	*0.05*	⇧
**Trajectory 5** Infected, without reinfections	**IgG(S)**, Log10 (BAU/mL), median [IQR]		**Neutralization Titers,** Log_10_ (reciprocal dilution), median [IQR]	
T5 month 0, all participants	2.865 [1.948, 3.447]	n = 18	1.842 [1.778,2.620]	n = 17
T5 month 3 all participants	3.447 [3.197, 3.950	n = 80	3.049 [2.650, 3.634]	n = 65
T5 month 9, all participants	3.563 [3.272, 3.898]	n = 69	2.741 [2.397, 3.074]	n = 56
T5 month 15, 3 doses	3.401 [2.827, 3.850]	n = 14	2.918 [1.778, 3.151]	n = 11
T5 month 15, 4 doses	3.876 [3.612, 4.260]	n = 52	2.908 [2.651, 3.560]	n = 41
*p-values, Wilcoxon*				
T5, Month 0 vs. month 3	*<0.001*	⇧	*<0.001*	⇧
T5, Month 3 vs. month 9	*0.78*	=	*0.01*	⇩
T5, Month 9 vs. month 15, 3 doses	*0.27*	=	*0.64*	=
T5, Month 9 vs. month 15, 4 doses	*<0.001*	⇧	*0.001*	⇧

**Table 3 vaccines-13-00738-t003:** Logistic regression analysis of post-vaccine infection at different time periods. Analysis of the risk of a new infection between 3 and 9 months after the administration of the third dose of the SARS-CoV-2 vaccine (A), between 3 and 15 months (B), and between the administration of the third dose and 15 months after (C), based on a previous infection and the levels of IgG(S) and neutralizing antibodies, adjusted for age, sex, and number of vaccine doses.

	A			B			C		
	3–9 Months (N = 197)			3–15 Months (N = 181)			0–15 Months (N = 76)		
	OR	95%CI	*p*-Value	OR	95%CI	*p*-Value	OR	95%CI	*p*-Value
Previous infection (*Ref*. uninfected)	0.077	[0.021; 0.213]	* **<0.001** *	0.043	[0.012; 0.120]	* **<0.001** *	0.048	[0.004; 0.276]	* **0.003** *
IgG(S) *	1.419	[0.446; 4.645]	*0.556*	1.691	[0.498; 5.969]	*0.404*	0.825	[0.340; 1.973]	*0.663*
Neutralizing antibodies *	0.638	[0.286; 1.391]	*0.263*	0.524	[0.216; 1.242]	*0.147*	3.224	[0.539; 24.94]	*0.195*
Age (*Ref*. 63–74 years)									
75–84	0.660	[0.293; 1.452]	*0.306*	0.483	[0.205; 1.111]	*0.089*	0.293	[0.075; 1.033]	*0.062*
>84	1.365	[0.550; 3.404]	*0.500*	1.128	[0.361; 3.664]	*0.838*	0		*0.990*
Sex (*Ref*. female)	0.850	[0.423; 1.700]	*0.645*	1.010	[0.472; 2.179]	*0.980*	1.084	[0.348; 3.574]	*0.891*
Vaccine doses (*Ref*. 3 doses)	-	-	-	0.950	[0.443; 2.036]	*0.896*	0.911	[0.316; 2.643]	*0.862*

** Log transformation.* Participants were stratified by prior infection status, with IgG(S), neutralizing antibodies, age, and sex as covariates. None of these variables significantly impacted reinfection risk during the 3–9 months post-third dose (Appendix A).

**Table 4 vaccines-13-00738-t004:** Cellular immune response differences by sex.

Sex	3-Month Visit		9-Month Visit		15-Month Visit	
	Female (n = 20)	Male (n = 11)	Female (n = 13)	Male (n = 11)	Female (n = 14)	Male (n = 10)
**Cellular immune response**						
Cellular response, pg/mL, *median* [Q1;Q3]	90.0 [36.1; 254]	42.8 [21.6; 76.2]	178 [120; 272]	33.8 [19.8; 71.2]	118 [72.2; 161]	47.9 [11.4; 87.4]
Cellular status, *n* (%)						
Positive	16 (80.0)	9 (81.8)	13 (100)	11 (100)	14 (100)	7 (70.0)
Negative	0 (0)	1 (9.1)	0 (0)	0 (0)	0 (0)	0 (0)
Undetermined	2 (10.0)	1 (9.1)	0 (0)	0 (0)	0 (0)	3 (30.0)
Not valuable	2 (10.0)	0 (0)	0 (0)	0 (0)	0 (0)	0 (0)

## Data Availability

The data used in this study are only available for the participating researchers, in accordance with current European and national laws. Thus, the distribution of the data is not allowed.

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
