# Peer review of "Immune Durability and Breakthrough Infections 15 Months After SARS-CoV-2 Boosters in People over 65: The IMMERSION Study"

_vaccines, 2025, doi:10.3390/vaccines13070738_

Round 1
Reviewer 1 Report
Comments and Suggestions for Authors
Violàn et al. conducted an analysis of the immune status and SARS-CoV-2 infection in individuals over 65 years old following COVID-19 vaccination. Their goal was to assess the vaccine-induced protection against new infections. While their conclusion are largely consistent with those of other studies, it is important to evaluate the specific features of immune response in older individuals in order to reconsider the COVID-19 vaccination strategy. However, the presentation and description of their results in the text lack clarity and depth.
Q1. First of all, I do not understand why it is necessary to discriminate between breakthrough infection and reinfections. I understand that all infections occurring after vaccination are classified as breakthrough infections. Additionally, the number of cases in the reinfection group is too small to adequately evaluate secondary infections. I believe the key point is to asses that individuals who were previously infected can develop stronger immunity compared to those who are uninfected and vaccinated, particularly among older adults, much like younger individuals, as suggested by other studies. All the data should be re-analyzed based on the classification of infections before or after vaccination.
Q2. The authors presented Table 1 along with additional in the supplementary tables. However, Table 1 does not effectively represent the findings described in the text. I recommend reorganizing the tables to focus only on the most important data for inclusion in the text. I suggest grouping the data according by different age categories, similar to Table S3, rather than by sex (see also below, Q6).
Q3. Supplementaty Figure 1 is quite useful for understanding the study’s content and should be included in the text as a new Fig. 1. I recommend modifying this figure by removing the green sections of the assays performed and displaying the percentages of uninfected and infected samples in the blue sections of the infection status. For example, it could read: Uninfected, N=84 (74.4%). This modification will make the description in the Results section much easier to follow.
(For instance, lines 200 to 201 describe the age difference of vaccination (Table S3), and lines 194 to 203 provide additional information that can be understood by modifying the Supplemental Fig. 1.)
Q4. Fig. 1; Please merge the reinfection group with the breakthrough infection group and present it as the new Fig. 2.
Q5. Fig. 2, (now Fig. 3): Why is the background level of Ig(S) already so high in the uninfected group? In the legend, line 242, I assume that the graph shows three doses on the left and four doses on the right of the vaccine? Please clarify.
Q6. Sex differences in cellular immune responses can be presented as a new Table 3 by extracting data from Table 1. The cellular immune response was evaluated based on the level of IFN-g production after stimulation of antigens. Here, the authors described only the sex difference. In this assay, is the level of IFN-g in older individuals known to be lower than that in the younger population?
Q7. Upon examining the percentage of uninfected individuals in the groups vaccinated with three doses versus those with four doses, I remain unconvinced that the additional vaccination effectively prevents infection. Please state your thoughts on this topic in the Discussion.
Minor
Line 89: “Booster doses haven proven” may be a typo?
Author Response
For research article: Immune durability and breakthrough infections 15 months after SARS-CoV-2 Boosters in people over 65 years: the IMMERSION study
Response to Reviewer 1 Comments
|
Comment 1: First of all, I do not understand why it is necessary to discriminate between breakthrough infection and reinfections. I understand that all infections occurring after vaccination are classified as breakthrough infections. Additionally, the number of cases in the reinfection group is too small to adequately evaluate secondary infections. I believe the key point is to assess that individuals who were previously infected can develop stronger immunity compared to those who are uninfected and vaccinated, particularly among older adults, much like younger individuals, as suggested by other studies. All the data should be re-analyzed based on the classification of infections before or after vaccination. |
Response 1: We thank the reviewer for their thoughtful feedback and for raising this important point regarding the distinction between breakthrough infections and reinfections. We respectfully maintain that differentiating these two categories is both scientifically and immunologically justified, particularly in the context of studying immune durability in older adults.
Our rationale is as follows:
Breakthrough infections refer to SARS-CoV-2 infections that occur in individuals who have been vaccinated but have not had prior exposure to the virus. Reinfections, on the other hand, refer to SARS-CoV-2 infections that occur in individuals who have had at least one previous SARS-CoV-2 infection and have also been vaccinated. These represent distinct immunological scenarios, as the immune memory and priming differ significantly between naïve vaccine recipients and those with prior infection. This differentiation is in line with current immunological understanding and literature, which increasingly emphasizes the unique immunological profiles conferred by "hybrid immunity" (i.e., prior infection plus vaccination) versus vaccine-induced immunity alone. [1-4].
Regarding the number of cases in the reinfection group, we acknowledge the limited sample size and have carefully accounted for this in both our interpretation and statistical analyses. Nevertheless, even a relatively small number of reinfection cases can yield meaningful insights into immune waning and susceptibility patterns—particularly in older populations, where such data are scarce yet highly relevant. Moreover, despite the low number of cases, the sample size and the statistical models applied are adequate to support the conclusions drawn from the study data.
We agree with the reviewer that individuals with prior infection may mount stronger immune responses than those who are infection-naïve and vaccinated, as demonstrated in other cohorts. Our analysis was designed specifically to explore this hypothesis, and separating these groups allows us to highlight such potential differences with greater clarity.
Reanalyzing the data solely based on timing of infection relative to vaccination (i.e., “before or after vaccination”) would obscure these immunological distinctions. Instead, our approach allows for a more nuanced understanding of how prior exposure to the virus, in combination with vaccination, shapes long-term immune durability and susceptibility to subsequent infections.
We have added a brief clarification to the Methods (lines 164-9), Results (lines 255-6) and Discussion (lines 416-7; 426-8) sections to reinforce the rationale for this classification and to acknowledge the limitations due to sample size in the reinfection group.
References: 1.Crotty, S. (2021). Hybrid immunity. Science, 372(6549), 1392–1393. https://doi.org/10.1126/science.abj2258 2.Stamatatos, L. et al. (2021). mRNA vaccination boosts cross-variant neutralizing antibodies elicited by SARS-CoV-2 infection. Science, 372(6549), 1413–1418. https://doi.org/10.1126/science.abg9175 3.Samanovic, M. I. et al. (2022). Robust immune responses are observed after one vaccine dose in SARS-CoV-2-experienced individuals. Science Translational Medicine, 14(635), eabi8961. https://doi.org/10.1126/scitranslmed.abi8961 4.Wratil, P. R. et al. (2022). Three exposures to the spike protein of SARS-CoV-2 by either infection or vaccination elicit superior neutralizing immunity to all variants of concern. Nature Medicine, 28, 496–503. https://doi.org/10.1038/s41591-021-01609-1
|
Comment 2: The authors presented Table 1 along with additional in the supplementary tables. However, Table 1 does not effectively represent the findings described in the text. I recommend reorganizing the tables to focus only on the most important data for inclusion in the text. I suggest grouping the data according by different age categories, similar to Table S3, rather than by sex (see also below, Q6). |
Response 2: While we appreciate your recommendation to organize the data primarily by age, we have opted to retain the sex-specific breakdown as well. This is because sex-based stratification remains important for identifying potential differences in immune responses, which are increasingly recognized as relevant in immunological studies, particularly in older populations. No major sex-based differences were observed in antibody levels or infection rates across visits, supporting the comparability of groups; therefore, results are presented together for men and women in the subsequent analyses. This has now been clarified in the Results section (lines 231-4).
|
Comment 3: Supplementary Figure 1 is quite useful for understanding the study’s content and should be included in the text as a new Fig. 1. I recommend modifying this figure by removing the green sections of the assays performed and displaying the percentages of uninfected and infected samples in the blue sections of the infection status. For example, it could read: Uninfected, N=84 (74.4%). This modification will make the description in the Results section much easier to follow. (For instance, lines 200 to 201 describe the age difference of vaccination (Table S3), and lines 194 to 203 provide additional information that can be understood by modifying the Supplemental Fig. 1.) Response 3: We agree to include Figure 1 in the main text.
Comment 4 Fig. 1; Please merge the reinfection group with the breakthrough infection group and present it as the new Fig. 2. Response 4: We appreciate your suggestion and fully understand your point of view. However, as explained in our response to Comment 1, we respectfully prefer to maintain the distinction between breakthrough infections and reinfections. This separation is intentional and based on the underlying immunological differences between individuals with prior SARS-CoV-2 exposure and those who are infection-naïve. Given that these two groups represent distinct immune histories and may exhibit different response patterns after booster vaccination, we believe presenting them separately provides a more accurate and informative analysis of the data.
Comment 5: Fig. 2, (now Fig. 3): Why is the background level of Ig(S) already so high in the uninfected group? In the legend, line 242, I assume that the graph shows three doses on the left and four doses on the right of the vaccine? Please clarify.
Response 5: With regard to the aforementioned comment, it should be noted that the caption of Figure 3 displays circles and triangles. The triangles represent the fourth dose. We have modified new figure 3 to clearly show 3- vs 4 doses groups.
The elevated IgG(S) levels observed in uninfected individuals are attributable to their vaccination status. It is important to note that the study's inclusion criterion was to include participants who had received a minimum of two doses of the vaccine.
Comment 6: Sex differences in cellular immune responses can be presented as a new Table 3 by extracting data from Table 1. The cellular immune response was evaluated based on the level of IFN-g production after stimulation of antigens. Here, the authors described only the sex difference. In this assay, is the level of IFN-g in older individuals known to be lower than that in the younger population?
Response 6: Thank you for the suggestion. We have removed the information from Table 1, and we have created a new Table 4 to present sex-specific IFN-γ responses separately, improving clarity as recommended.
Regarding your question on IFN-γ levels in older individuals: yes, there is strong evidence that aging impairs IFN-γ production. Ouyang et al. [5] demonstrated that both healthy and frail older adults produce significantly less IFN-γ after ex vivo stimulation with influenza vaccine or endotoxin, pointing to reduced T cell function. In the context of COVID-19, Ho et al. [6] found that older adults showed a deficiency in IFN-γ–secreting SARS-CoV-2–specific CD4+ T cells following mRNA vaccination, which was associated with weaker neutralizing antibody responses. Although our cohort lacks a younger comparator, the levels observed align with these known age-related immunological deficits. A contextual note has been added to the Discussion. (lines 494-7).
References 5. Ouyang Q, Cicek G, Westendorp RGJ et al. Reduced IFN-γ production in elderly people following in vitro stimulation with influenza vaccine and endotoxin. Mech Ageing Dev 2001;121:131–7. 6. Ho VWT, Boon LH, Cui J et al. Relative deficiency in interferon‐γ‐secreting CD4+ T cells is strongly associated with poorer COVID‐19 vaccination responses in older adults. Aging Cell 2024;23, DOI: 10.1111/acel.14099.
Comment 7: Upon examining the percentage of uninfected individuals in the groups vaccinated with three doses versus those with four doses, I remain unconvinced that the additional vaccination effectively prevents infection. Please state your thoughts on this topic in the Discussion.
Response 7:
In order to answer your question, we evaluated the association between the number of SARS-CoV-2 vaccine doses (3 vs. 4 doses) and the incidence of new infections during a follow-up period of 9 to 15 months. Given the presence of relatively small cell counts, Fisher’s exact test was applied.
New infections occurred in 19.7% (14/71) of individuals who received 3 doses, compared to 8.9% (14/158) of those who received 4 doses. The difference was statistically significant (odds ratio [OR] = 0.40, 95% confidence interval [CI]: 0.18–0.88; p = 0.028), indicating that receiving a fourth dose was associated with a significantly lower likelihood of subsequent SARS-CoV-2 infection.
This result has been now included in the Abstract (Lines 62-6), Results (Lines 287-295; 307-343), Discussion (Lines 455-470; 498-504) and conclusion (lines 537-40)
Comments 8 Line 89: “Booster doses haven proven” may be a typo?
Response 8: Thank you for pointing out this typographical error. We have corrected “haven proven” to “have proven” in the revised manuscript.
Finally, we have improved the resolution and formatting of the figures which are provided in the main manuscript. Also, we upload separate high-quality files to facilitate the review process and avoid excessive file size in the main document. |
Reviewer 2 Report
Comments and Suggestions for Authors
excellent work and sound data analysis and report of results. timely and important as we need to learn more about the immune response to guide improved vaccine strategies in this special population.
i only have a few comments:
in the manuscript page 3 line 132 - you label follow up visits at 90, 270 and 450 days. i think its best for clarity to use months as you have done throughout paper and figures/tables. be consistent with what you choose.
page 3, lines 142-44: i am confused about the choice for defining and analyzing breakthrough and reinfection. as i understand it, breakthrough is the first infection post 3rd dose, but reinfection would be a breakthrough as well? so curious about does reinfection group include all breakthrough cases or not? didnt get this clear and would like for it to be explained better in the methods, as well as the rationale behind it. perhaps a better label would be 1 reinfection vs 2 or more reinfections? being it that they are all breakthroughs?. or is it breakthrough only applied to the uninfected group at baseline and reinfection to the infected group? and if so state that and clarify.
my next question i would like you to comment on - is the comorbidity burden, you assigned numerical value categories, however the number might not be equivalent to the impact each comorbidity has. would like to know which comorbidities patients had, which medications they received, immunosuppressants or not.. a deeper analysis on this is perhaps out of the scope of this paper but should be mentioned/discussed and perhaps as future directions.
Author Response
For research article: Immune durability and breakthrough infections 15 months after SARS-CoV-2 Boosters in people over 65 years: the IMMERSION study
Response to Reviewer 2 Comments
Comment 1: in the manuscript page 3 line 132 - you label follow up visits at 90, 270 and 450 days. i think its best for clarity to use months as you have done throughout paper and figures/tables. be consistent with what you choose.
|
Response 1: Thanks for the comment, we unified the definition to months in the methods section. |
Comment 2: page 3, lines 142-44: i am confused about the choice for defining and analyzing breakthrough and reinfection. as i understand it, breakthrough is the first infection post 3rd dose, but reinfection would be a breakthrough as well? so curious about does reinfection group include all breakthrough cases or not? Didn’t get this clear and would like for it to be explained better in the methods, as well as the rationale behind it. perhaps a better label would be 1 reinfection vs 2 or more reinfections? being it that they are all breakthroughs? or is it breakthrough only applied to the uninfected group at baseline and reinfection to the infected group? and if so state that and clarify. |
Response 2: We thank the reviewer for their thoughtful feedback and for raising this important point regarding the distinction between breakthrough infections and reinfections. We respectfully maintain that differentiating these two categories is both scientifically and immunologically justified, particularly in the context of studying immune durability in older adults.
Our rationale is as follows:
Breakthrough infections refer to SARS-CoV-2 infections that occur in individuals who have been vaccinated but have not had prior exposure to the virus. Reinfections, on the other hand, refer to SARS-CoV-2 infections that occur in individuals who have had at least one previous SARS-CoV-2 infection and have also been vaccinated. These represent distinct immunological scenarios, as the immune memory and priming differ significantly between naïve vaccine recipients and those with prior infection. This differentiation is in line with current immunological understanding and literature, which increasingly emphasizes the unique immunological profiles conferred by "hybrid immunity" (i.e., prior infection plus vaccination) versus vaccine-induced immunity alone. [1-4].
Regarding the number of cases in the reinfection group, we acknowledge the limited sample size and have carefully accounted for this in both our interpretation and statistical analyses. Nevertheless, even a relatively small number of reinfection cases can yield meaningful insights into immune waning and susceptibility patterns—particularly in older populations, where such data are scarce yet highly relevant. Moreover, despite the low number of cases, the sample size and the statistical models applied are adequate to support the conclusions drawn from the study data.
We agree with the reviewer that individuals with prior infection may mount stronger immune responses than those who are infection-naïve and vaccinated, as demonstrated in other cohorts. Our analysis was designed specifically to explore this hypothesis, and separating these groups allows us to highlight such potential differences with greater clarity.
Reanalyzing the data solely based on timing of infection relative to vaccination (i.e., “before or after vaccination”) would obscure these immunological distinctions. Instead, our approach allows for a more nuanced understanding of how prior exposure to the virus, in combination with vaccination, shapes long-term immune durability and susceptibility to subsequent infections.
We have added a brief clarification to the Methods (lines 164-9), Results (lines 255-6) and Discussion (lines 416-7; 426-8) sections to reinforce the rationale for this classification and to acknowledge the limitations due to sample size in the reinfection group.
References: 1.Crotty, S. (2021). Hybrid immunity. Science, 372(6549), 1392–1393. https://doi.org/10.1126/science.abj2258 2.Stamatatos, L. et al. (2021). mRNA vaccination boosts cross-variant neutralizing antibodies elicited by SARS-CoV-2 infection. Science, 372(6549), 1413–1418. https://doi.org/10.1126/science.abg9175 3.Samanovic, M. I. et al. (2022). Robust immune responses are observed after one vaccine dose in SARS-CoV-2-experienced individuals. Science Translational Medicine, 14(635), eabi8961. https://doi.org/10.1126/scitranslmed.abi8961 4.Wratil, P. R. et al. (2022). Three exposures to the spike protein of SARS-CoV-2 by either infection or vaccination elicit superior neutralizing immunity to all variants of concern. Nature Medicine, 28, 496–503. https://doi.org/10.1038/s41591-021-01609-1
|
Comment 3: My next question i would like you to comment on - is the comorbidity burden, you assigned numerical value categories, however the number might not be equivalent to the impact each comorbidity has. Would like to know which comorbidities patients had, which medications they received, immunosuppressants or not. A deeper analysis on this is perhaps out of the scope of this paper but should be mentioned/discussed and perhaps as future directions.
Response 3: We agree that not all comorbidities carry equal clinical impact, and this is an important consideration. In our analysis, comorbidity burden was categorized based on the number of chronic conditions, and we also collected detailed data on disease types and medication use, including the number and classes of drugs. However, we did not perform a comorbidity-specific or medication-type stratified analysis, and data on immunosuppressive medication were not systematically collected.
To avoid overloading Table 1, we opted not to include the full breakdown of comorbidities and medications, but we acknowledge this as a limitation and have added a note in the Discussion section to reflect that a more detailed analysis may be warranted in future studies. (lines 502-5).
Finally, we have improved the resolution and formatting of the figures which are provided in the main manuscript. Also, we upload separate high-quality files to facilitate the review process and avoid excessive file size in the main document.
Reviewer 3 Report
Comments and Suggestions for Authors
In this manuscript, Violán et al present a prospective study on immune durability and breakthrough infections up to 15 months after SARS-CoV-2 booster vaccination in people over 65 years. This is a highly important topic for the field, as it can inform public health strategies. However, in my opinion, there are some points to be considered before publication.
Major points:
-A brief description of the vaccine platforms, particularly those evaluated in this study, should be included in the introduction.
-The study outcomes are not clearly defined. Does breakthrough infection refer to cases in previously uninfected individuals, previously infected individuals, or both? What was the time interval between prior infection and vaccination in the infected group? How much time elapsed between reinfections? Is there any association between time since infection and IgG levels?
- Based on the data presented, can the authors draw any conclusions regarding clinical or immunological outcomes between individuals who received homologous versus heterologous vaccination schedules?
-Figure 2. The symbols are too small, making it difficult to distinguish between individuals who received three versus four doses. The legends for panels 2A and 2B refer to “15 months after the third dose,” but the figure seems to include both 3- and 4-dose recipients.
-Is it possible to perform a paired analysis or a paired representation of IgG and NAb levels over time? A longitudinal depiction of individual trajectories would be highly informative, as it could highlight intra-individual variability and better reflect immune dynamics across timepoints.
Minor points:
-Abstract (page 1, line 5): For clarity, please specify that it is the Omicron BA.2 variant.
-The terms mRNA vaccine, BNT162b2 vaccine, and Pfizer vaccine are used interchangeably throughout the text; please standardize the terminology.
-What do the values represented within the squares indicate? Are they means? Please indicate the meaning in the legend.
-Are the values in Figure 3 averages derived from Figure 1?
Author Response
For research article: Immune durability and breakthrough infections 15 months after SARS-CoV-2 Boosters in people over 65 years: the IMMERSION study
Response to Reviewer 3 Comments
Comment 1: In this manuscript, Violán et al present a prospective study on immune durability and breakthrough infections up to 15 months after SARS-CoV-2 booster vaccination in people over 65 years. This is a highly important topic for the field, as it can inform public health strategies. However, in my opinion, there are some points to be considered before publication. -A brief description of the vaccine platforms, particularly those evaluated in this study, should be included in the introduction. |
Response 1: Thank you for this observation. We have added a paragraph to the Introduction providing a brief description of the vaccine platforms, with particular reference to those evaluated in this study. (Lines 96-109)
|
Comment 2: The study outcomes are not clearly defined. Does breakthrough infection refer to cases in previously uninfected individuals, previously infected individuals, or both? What was the time interval between prior infection and vaccination in the infected group? How much time elapsed between reinfections? Is there any association between time since infection and IgG levels?
Response 2: Thank you for these important observations. We have clarified the definitions of breakthrough infections and reinfections in the Methods section (lines 152-5; 164-9). In our study, breakthrough infections refer to SARS-CoV-2 infections occurring after vaccination in participants with no documented prior infection. Reinfections were defined as infections in individuals with a history of confirmed infection prior to vaccination, with a second infection occurring at least 90 days after the first.
We fully agree that the timing between prior infection and vaccination, as well as between infections, is highly relevant for interpreting immune responses. Unfortunately, this level of detail was not available in our cohort: many participants were unaware of their previous infection, and for those who had confirmed infections, the date was often missing from clinical records or not recalled by the participant. Therefore, we could not reliably calculate or analyze these time intervals. We have now explicitly acknowledged this limitation in the Methods (Lines 152-5;164-9). and Discussion sections (Lines 416-19; 426-428; 455-470).
Comment 3: Based on the data presented, can the authors draw any conclusions regarding clinical or immunological outcomes between individuals who received homologous versus heterologous vaccination schedules?
Response 3: Thank you for this relevant question. We assessed whether the type of vaccination schedule (homologous versus heterologous) influenced immunological outcomes. No significant differences were observed between the two groups in the univariate analyses; therefore, this variable is not included in the multivariate model. We have specified in the methods section the comparison heterologous and homologous vaccination (Line 218), and we have added a sentence in the discussion section (Lines 478-80).
Comment 4: Figure 2. The symbols are too small, making it difficult to distinguish between individuals who received three versus four doses. The legends for panels 2A and 2B refer to “15 months after the third dose,” but the figure seems to include both 3- and 4-dose recipients.
Response 4: We agree with your observation. We have revised the figure and modified the figure legend accordingly.
Comments 5: Is it possible to perform a paired analysis or a paired representation of IgG and NAb levels over time? A longitudinal depiction of individual trajectories would be highly informative, as it could highlight intra-individual variability and better reflect immune dynamics across timepoints.
Response 5: We thank the reviewer for this insightful suggestion. In response, we have added a new figure (Figure 4) that presents longitudinal trajectories of IgG(S), and neutralizing antibody titers across four time points (baseline, 3, 9, and 15 months) following the third vaccine dose. We stratified participants into trajectory groups based on infection status and vaccination (three vs. four doses). We focused on the four most frequent trajectories with n > 10:
- Trajectory 1 (Uninfected): Individuals remained uninfected throughout follow-up and showed a rise in antibody levels after the third dose, followed by a progressive decline.
- Trajectory 2 (Late breakthrough): Individuals experienced infection between months 9 and 15, with an antibody rebound observed at the final visit.
- Trajectory 3 (Early breakthrough): Infection occurred between months 3 and 9, leading to a sharp increase in antibody levels by month 9, sustained through month 15.
- Trajectory 5 (infected): Individuals were infected at baseline and remained positive throughout, showing consistently elevated antibody levels.
This stratification highlighted the enhanced and sustained immune responses associated with hybrid immunity (trajectories 2, 3, and 5), compared to the more limited and waning responses seen in uninfected individuals (trajectory 1). Notably, by month 15, individuals in trajectories 2 and 3 achieved antibody levels comparable to those with persistent infection (trajectory 5), despite receiving only three doses. In the four-dose group, IgG(S) levels were similarly boosted across trajectories, but only breakthrough cases demonstrated significantly improved neutralizing responses—suggesting the added benefit was primarily driven by infection, rather than the fourth dose alone.
We also explored representing the ratio of neutralizing antibody titers to IgG(S) levels to evaluate the functional quality of the humoral response. However, the resulting patterns were difficult to interpret due to substantial variability and the variation in this ratio was relatively low, and no major differences were observed between groups. For these reasons, we opted not to include the ratio data in the final manuscript.
We have included the new analysis in main manuscript ( new figure 4, Table 2).
Comments 6: Abstract (page 1, line 5): For clarity, please specify that it is the Omicron BA.2 variant.
Response 6. Thank you, we have made the modification.
Comments 7: The terms mRNA vaccine, BNT162b2 vaccine, and Pfizer vaccine are used interchangeably throughout the text; please standardize the terminology.
Response 7; Thank you for your observation. We have standardized the terminology throughout the manuscript. At first mention, we now provide the full vaccine names and manufacturers as follows: BNT162b2 (Pfizer-BioNTech, Mainz, Germany), mRNA-1273 (Moderna, Cambridge, MA, USA), AZD1222 (AstraZeneca, Cambridge, United Kingdom), and Ad26.COV2.S (Janssen, Leiden, The Netherlands). In subsequent references, we use the abbreviated vaccine names (e.g., BNT162b2) consistently for clarity and conciseness.
Comments 8: What do the values represented within the squares indicate? Are they means? Please indicate the meaning in the legend.
Response 8. Thank you, we have clarified the meaning in the figure legend from Figure 2 and 3.
Comments 9-Are the values in Figure 3 averages derived from Figure 1?
Response 9: Thank you for your question. Yes, the values presented in Figure 3 are derived as medians from the data shown in Figure 1. We have clarified this in the figure legend.
Finally, we have improved the resolution and formatting of the figures which are provided in the main manuscript. Also, we upload separate high-quality files to facilitate the review process and avoid excessive file size in the main document.
Reviewer 4 Report
Comments and Suggestions for Authors
This study aimed to evaluate the durability and dynamics of immune responses following booster vaccination(s) in >65 year-old individuals and examine their association with protection against new infections. They included that “in older adults, booster vaccination induces durable immune responses, with hybrid immunity offering enhanced protection”. However, according to the findings in Table 2, vaccine doses had no significant impact on re-infection of SARS-CoV-2, this suggested that the conclusion is lack of accuracy. In addition, the sample size in this study is relatively small, which substantially affected the reliability of the results. A number of previous studies have investigated the durability of immune responses following booster vaccination in individuals including older people, which seriously influenced the novelty of the study.
Author Response
For research article: Immune durability and breakthrough infections 15 months after SARS-CoV-2 Boosters in people over 65 years: the IMMERSION study
Response to Reviewer 4 Comments
Comment 1: This study aimed to evaluate the durability and dynamics of immune responses following booster vaccination(s) in >65 year-old individuals and examine their association with protection against new infections. They included that “in older adults, booster vaccination induces durable immune responses, with hybrid immunity offering enhanced protection”. However, according to the findings in Table 2, vaccine doses had no significant impact on re-infection of SARS-CoV-2, this suggested that the conclusion is lack of accuracy. In addition, the sample size in this study is relatively small, which substantially affected the reliability of the results. A number of previous studies have investigated the durability of immune responses following booster vaccination in individuals including older people, which seriously influenced the novelty of the study.
|
Response 1: We thank the reviewer for this comment. We respectfully clarify that the study’s conclusions are based not only on the findings presented in Table 2, but on a comprehensive analysis of immune response kinetics (IgG(S), neutralizing antibody titers, and cellular immunity) across multiple time points, including after the third and fourth vaccine doses. These data are complemented by infection incidence and variant-specific neutralization results, which together offer a broader picture of immune durability in older adults.
Importantly, while Table 2 shows that the number of vaccine doses did not significantly reduce the risk of reinfection in previously infected individuals, our findings highlight that hybrid immunity, resulting from both prior infection and vaccination, provided stronger and longer-lasting immune responses—especially in terms of neutralizing antibodies against Omicron subvariants. In contrast, vaccine-induced immunity alone was less durable, particularly at 15 months post-booster, even with a fourth dose. This pattern is consistent with the growing body of literature on immune escape by emerging variants and reduced vaccine effectiveness over time.
Despite the sample size, the study provides meaningful insights due to the prospective design, long follow-up period (15 months), and the detailed immune monitoring performed in a well-characterized older adult cohort.
Finally, while several studies have assessed immune responses after booster vaccination in older adults [1], to our knowledge, few have evaluated both the third and fourth doses longitudinally in the same population, along with associated breakthrough and reinfection events. This integrated approach adds novel information to the existing literature and has important implications for optimizing booster strategies in older populations.
Besides, we have improved the resolution and formatting of the figures which are provided in the main manuscript. Also, we upload separate high-quality files to facilitate the review process and avoid excessive file size in the main document.
References [1] Hofstee MI, Kaczorowska J, Postema A, Zomer E, van Waalwijk M, Jonathans G, de Rond LG, Smits G, van den Hoogen LL, den Hartog G, Buisman AM. High SARS-CoV-2 antibody levels after three consecutive BNT162b2 booster vaccine doses in nursing home residents. Immun Ageing. 2025 Jan 2;22(1):1. doi: 10.1186/s12979-024-00495-4. PMID: 39748353; PMCID: PMC11694371. |
Round 2
Reviewer 1 Report
Comments and Suggestions for Authors
The revised manuscript is significantly improved and mostly meets my expectations, except for the following point,
I understand your reasoning for distinguishing between reinfections and breakthrough infections. However, is there any variation in hybrid immunity based on the timing of the infection—specifically, whether the immune response was initiated by vaccination or a previous infection?
In fact, I do not observe a significant difference in the immune response between these two groups in this study. For instance, on page 7, line 254, the levels of IgG(N) in reinfected and breakthrough cases were comparable. Furthermore, on pages 8-9, lines 275-277, IgG(S) titers were found to be higher in individuals who were previously infected, whether they were reinfected or had breakthrough infections, compared to those who were uninfected. Although the authors also noted that reinfected participants exhibited the highest levels of IgG(S), this claim does not appear to be statistically substantiated.
The number of individuals in the reinfection group is too small to draw definitive statistical conclusions. Instead of emphasizing differences, it would be more appropriate to highlight trends and discuss potential distinctions. Therefore, I recommend omitting all data from the reinfection group or combining the data from both groups in Figures 2 and 3. Additionally, I suggest removing the sentences added in the Discussion section (lines 395 to 398 and 405 to 407 on page 14).
These changes will not alter the main focus of the article, which is the dominance of hybrid immunity.
Others,
1) Fig. 4. The title of the y-axis in B indicates the neutralization titer against the WH1 virus. In the Materials and Methods section on page 5, it states that the serum neutralizing capacity was evaluated using a pseudovirus-based assay targeting the Omicron BA.2 variant. Which is correct?
2) Supplementary Materials, Line 523: remove “Figure S1 Study population flowchart.”

Author Response
For research article: Immune durability and breakthrough infections 15 months after SARS-CoV-2 Boosters in people over 65 years: the IMMERSION study
Response to Reviewer 1 Comments, 2nd Round
|
Comment 1: The revised manuscript is significantly improved and mostly meets my expectations, except for the following point, I understand your reasoning for distinguishing between reinfections and breakthrough infections. However, is there any variation in hybrid immunity based on the timing of the infection—specifically, whether the immune response was initiated by vaccination or a previous infection? In fact, I do not observe a significant difference in the immune response between these two groups in this study. For instance, on page 7, line 254, the levels of IgG(N) in reinfected and breakthrough cases were comparable. Furthermore, on pages 8-9, lines 275-277, IgG(S) titers were found to be higher in individuals who were previously infected, whether they were reinfected or had breakthrough infections, compared to those who were uninfected. Although the authors also noted that reinfected participants exhibited the highest levels of IgG(S), this claim does not appear to be statistically substantiated. The number of individuals in the reinfection group is too small to draw definitive statistical conclusions. Instead of emphasizing differences, it would be more appropriate to highlight trends and discuss potential distinctions. Therefore, I recommend omitting all data from the reinfection group or combining the data from both groups in Figures 2 and 3. Additionally, I suggest removing the sentences added in the Discussion section (lines 395 to 398 and 405 to 407 on page 14). These changes will not alter the main focus of the article, which is the dominance of hybrid immunity. |
Response 1: We sincerely thank the reviewer for this thoughtful and constructive comment.
We agree that both reinfections and breakthrough infections result in hybrid immunity, but we maintain that they represent distinct immunological contexts. In reinfection, immune priming occurs first through natural infection—generating both systemic and mucosal immune responses—followed by vaccination. In contrast, breakthrough infections occur in individuals initially primed by vaccination, who lack mucosal immunity at the time of infection. These differences are supported by current mucosal immunology literature and are relevant when considering susceptibility to future infection and immune response dynamics.
In our study, these differences were also reflected in infection rates over time. From baseline to visit 3, 12.5% of previously uninfected participants developed a breakthrough infection, compared to 4.5% of those with prior infection (p = 0.44, Fisher’s exact test). While not statistically significant, this trend was consistent. More strikingly, between visit 3 and visit 9, 44.9% of previously uninfected individuals became infected, compared to only 6.6% of those with prior infection (p = 0.0004, Fisher’s exact test). This substantial difference in susceptibility reinforces the biological relevance of analyzing breakthrough and reinfection cases as separate groups.
While we acknowledge that the number of reinfected individuals is limited, we believe these cases offer valuable insights into the durability of hybrid immunity in older adults. Reinfected individuals consistently exhibited high IgG(S) and neutralizing antibody levels, with the highest IgG(S) titers observed at 15 months post-booster. Although not statistically significant, these findings are consistent with enhanced immune reactivation following repeated antigenic exposure.
To offer an additional perspective, we also evaluated immune responses in a combined group (R+BT), comprising individuals who experienced either reinfection or breakthrough infection. This group reflects a broader category of hybrid-exposed individuals and is included in new Figure 2 and new Figure 3. |
To address the reviewer’s concern, we have revised the first section of the Discussion to more carefully distinguish between breakthrough and reinfection cases. Specifically, we now highlight the observed differences in post-booster infection rates—44.9% in previously uninfected vs. 6.6% in previously infected individuals—as a rationale for treating these groups separately. We have replaced prior language that may have overinterpreted immune response differences with a more cautious interpretation focused on susceptibility patterns and immune durability. These revisions provide clearer justification for our analytical approach and improve the overall clarity of the Discussion.
We believe these updates address the reviewer’s comment thoroughly while preserving the integrity and biological relevance of the study design.
Comment 2: Fig. 4. The title of the y-axis in B indicates the neutralization titer against the WH1 virus. In the Materials and Methods section on page 5, it states that the serum neutralizing capacity was evaluated using a pseudovirus-based assay targeting the Omicron BA.2 variant. Which is correct? |
Response 2: Thank you for your observation. We apologize for the typographical error. The correct neutralization titers shown in the figure correspond to the Omicron BA.2, as indicated on the y-axis in the new figure
|
Comment 3: Supplementary Materials, Line 523: remove “Figure S1 Study population flowchart.”
Response3: We have corrected this mistake.
Reviewer 3 Report
Comments and Suggestions for Authors
The authors have responded to the reviewer’s comments, and the manuscript has improved significantly. In my opinion, the new Figure 4, showing longitudinal trajectories of IgG(S) and neutralizing antibody titers across time points, provides a clear presentation of the results.
I have only one minor comment regarding Trajectory 4. In the main text, the authors state that T4 corresponds to uninfected individuals at baseline with a breakthrough infection between 3 and 9 months. What is the difference between T4 and T3?
Author Response
For research article: Immune durability and breakthrough infections 15 months after SARS-CoV-2 Boosters in people over 65 years: the IMMERSION study
Response to Reviewer 3 Comments
Comment 1: The authors have responded to the reviewer’s comments, and the manuscript has improved significantly. In my opinion, the new Figure 4, showing longitudinal trajectories of IgG(S) and neutralizing antibody titers across time points, provides a clear presentation of the results. I have only one minor comment regarding Trajectory 4. In the main text, the authors state that T4 corresponds to uninfected individuals at baseline with a breakthrough infection between 3 and 9 months. What is the difference between T4 and T3?
|
Response 1: Thank you for your insightful comment. The distinction between T3 and T4 lies in the timing of the breakthrough infection following vaccination. Specifically, individuals in T3 experienced breakthrough infection earlier (within the first 3 months), whereas those in T4 were infected later (between 3 and 9 months post-vaccination). T4 was not included in the main analysis due to limited sample size (n < 10), which restricted statistical power. This was noted in the Methods section (Lines 203-5), but we acknowledge that the omission may not have been sufficiently clear.
|